# Hierarchical Filtering and Refinement Classification for Few-Shot Class-Incremental Learning

**Li-Jun Zhao, Zhen-Duo Chen**\*, **Xin Luo, Xin-Shun Xu**
*School of Software, Shandong University, China*
*lj_zhao1028@163.com, chenzd.sdu@gmail.com, luoxin.lxin@gmail.com, xuxinshun@sdu.edu.cn*

**Reviewed on OpenReview:** *https://openreview.net/forum?id=7MXra1JSh8*

## Abstract

Few-shot class-incremental learning (FSCIL) aims at recognizing novel classes continually with limited novel class samples. A mainstream baseline for FSCIL is first to train the whole model in the base session, then freeze the feature extractor in the incremental sessions. Despite achieving high overall accuracy, most methods exhibit notably low accuracy on incremental classes. While some recent methods have recognized this issue, their strategies remain constrained by a unified classification objective across all samples, making it difficult to simultaneously satisfy the performance requirements of both base and incremental classes. In this paper, considering that base and incremental classes play different yet both critical roles in FSCIL, we approach FSCIL from a more structured perspective by decomposing the overall classification objective into three sub-objectives. Building on this insight, we propose a novel classification framework called Hierarchical Filtering and Refinement Classification (HFRC) to hierarchically decompose and address the classification task. Extensive experiments demonstrate that our method effectively balances the classification accuracy between base and incremental classes, and achieves superior performance compared to state-of-the-art methods. Codes are available at: `https://github.com/Legenddddd/HFRC`.

## 1 Introduction

In the dynamic and open real world, Class-Incremental Learning (CIL) (Rebuffi et al., 2017; Hou et al., 2019) is proposed to continuously learn new emerging concepts and not forget the learned ones. However, humans can establish new concepts with only a few new examples when they have a certain amount of knowledge. Therefore, Few-Shot Class-Incremental Learning (FSCIL) (Gidaris & Komodakis, 2018; Tao et al., 2020) is proposed to continuously learn novel classes with limited novel class samples after training on base classes with sufficient samples.

In contrast to CIL which has the same number of classes to be learned in each session, FSCIL involves a significantly larger number of classes in the base session (base classes) compared to the incremental sessions. This problem setup allows the model to learn a large amount of knowledge during the base session, making it possible to learn novel classes (incremental classes) with limited training samples. Therefore, most FSCIL methods (Zhang et al., 2021; Hersche et al., 2022) decouple the learning of representations and classifiers, train the whole model in the base session, then freeze the feature extractor and only optimize classifiers in the incremental sessions. This strategy significantly alleviates catastrophic forgetting and overfitting in FSCIL, leading to notable improvements in overall classification accuracy. However, it exhibits particularly low accuracy on incremental classes.

Recent research (Wang et al., 2023b; Cui et al., 2024; Hu et al., 2025b; Zhao et al., 2025) has recognized this phenomenon and attempted to analyze and address it. These methods either focus on enhancing the

---

\*Corresponding author

learning and classification ability for incremental classes or aim to better distinguish between base and incremental classes. Although these methods have contributed to narrowing the performance gap between base and incremental classes, their strategies remain confined to a unified classification objective across all samples, resulting in difficulties to simultaneously accommodate the performance requirements of both base and incremental classes. On the one hand, base classes support feature learning for incremental classes, with the expectation that incremental classes can be accurately recognized. On the other hand, base classes require accurate classification themselves and may potentially interfere with the classification of incremental classes during inference. Thus, considering that base and incremental classes play different yet both critical roles in FSCIL, it is beneficial to approach classification with a more structured perspective. **Using the chain rule of probability, we decompose the overall classification objective into three sub-objectives: classification between base and incremental classes, intra-group classification within base classes, and intra-group classification within incremental classes.** Tackling these sub-objectives individually makes it easier to balance the performance between base and incremental classes, thus providing a more comprehensive solution to FSCIL.

In this paper, to decompose the overall classification objective into sub-objectives and address them separately, we propose a novel classification framework, **Hierarchical Filtering and Refinement Classification (HFRC)**, which balances the classification accuracy between base and incremental classes. Specifically, we introduce a selection and reorganization module after the feature extractor to enable hierarchical feature extraction. This module preserves transferable features that benefit incremental classes and subsequently extracts class-specific features tailored for base classes, thereby hierarchically decomposing the classification task. During classification, transferable features are first utilized to filter between base and incremental classes and to perform classification within incremental classes. For samples identified as belonging to base classes, class-specific features are further employed to refine the base-vs-incremental decision, particularly for samples similar to the base classes, and to conduct finer-grained classification within the base classes. In addition, we design a targeted classifier adjustment mechanism to further balance the accuracy between base and incremental classes.

Our key contributions are summarized as follows:

- We approach FSCIL from a more structured perspective by decomposing the overall classification objective into three sub-objectives: classification between base and incremental classes, intra-group classification within base classes, and intra-group classification within incremental classes.

- To tackle these classification sub-objectives, we propose a comprehensive solution named Hierarchical Filtering and Refinement Classification (HFRC). It consists of hierarchical feature extraction, which learns two complementary types of features during feature learning, and filtering and refinement classification, which exploits these features to address the decomposed classification sub-objectives during inference.

- Extensive experiments on multiple benchmark datasets demonstrate that our method effectively balances the classification accuracy between base and incremental classes, achieving consistently superior performance over state-of-the-art methods.

## 2 Related Works

### 2.1 Few-Shot Learning

Few-Shot Learning (FSL) (Qi et al., 2018; Lafargue et al., 2024) aims to quickly adapt to novel classes with limited training samples. Existing works can be broadly divided into two categories. Metric-based methods (Vinyals et al., 2016; Snell et al., 2017; Zhao et al., 2024) focus on learning discriminative embedding spaces where novel classes can be effectively recognized with nearest-neighbor or prototype-based strategies. Optimization-based methods (Finn et al., 2017; Ye & Chao, 2022; Zhang et al., 2022), on the other hand, aim to learn models or initialization strategies that can be rapidly adapted to new tasks through fine-tuning. These methods demonstrate that explicitly modeling pairwise or higher-order dependencies via graph structures can further enhance recognition ability, showing the importance of capturing cross-sample

or cross-class interactions. Nevertheless, all of the above methods remain confined to the FSL setting, which only concerns performance on novel classes. They do not address the continual learning challenge in Few-Shot Class-Incremental Learning (FSCIL), where both base and novel classes must be jointly considered.

## 2.2 Class-Incremental Learning

In Class-Incremental Learning (CIL) (Rolnick et al., 2019; Hou et al., 2019), the key challenge is to acquire novel classes while retaining knowledge of previously learned ones. Existing studies can be broadly categorized into three lines of research. Firstly, regularization-based methods (Zenke et al., 2017; Lin et al., 2023) attempt to alleviate forgetting by identifying parameters that are crucial for old classes and restricting their update during the learning of new classes. Secondly, data-replay methods (Chaudhry et al., 2019; Zhao et al., 2020; Hu et al., 2021) mitigate catastrophic forgetting by storing and replaying exemplars of previously encountered classes. Lastly, model expansion approaches (Yan et al., 2021; Wang et al., 2023a) dynamically allocate new sub-networks or expand model capacity to accommodate novel classes. Despite the progress in these three directions, existing CIL approaches assume access to sufficient labeled data for each class and do not consider scenarios where newly introduced classes have limited labeled samples.

## 2.3 Few-Shot Class-Incremental Learning

Few-shot Class-Incremental Learning (FSCIL) (Gidaris & Komodakis, 2018; Roy et al., 2024; Zhao et al., 2026) aims to address the challenge of CIL in scenarios with insufficient labeled data. Most FSCIL methods (Zhang et al., 2021; Hersche et al., 2022) involve freezing the parameters of the feature extractor in the incremental sessions and recognizing novel classes through prototype-based Nearest-Neighbor classification. However, using a feature extractor trained solely on base classes for incremental classes leads to significantly lower accuracy on incremental classes.

Recently, several methods have noticed this phenomenon and attempted to improve the accuracy of incremental classes. Specifically, TEEN (Wang et al., 2023b) calibrates the classifiers for incremental classes by fusing incremental prototypes with weighted base prototypes. OSHHG (Cui et al., 2024) introduces two learning modules based on hyperbolic geometry to enhance the model's ability to learn incremental classes. CSR (Hu et al., 2025b) expands the feature distributions of incremental classes by assigning a fixed covariance and pushes incremental class samples away from semantically similar distributions. D2A (Zhao et al., 2025) improves incremental class performance by diminishing and distributing the attraction of base class prototypes from the perspectives of both the distance metric and the feature space.

In contrast to prior methods, this paper approaches the classification process itself and proposes a novel method that more directly and effectively balances the accuracy between base and incremental classes. To this end, we explicitly analyze the influence of the fully connected module and identify two types of features that are critical for FSCIL classification, which form the foundation of our classification design. Although some existing FSCIL methods (Zou et al., 2022; Song et al., 2023) also adopt the fully connected module, they either leverage its position in the network to introduce additional loss terms or use it to construct contrastive learning frameworks. However, they neither recognize its inherent influence on feature representations and the FSCIL task itself, nor leverage it to achieve a breakthrough in balancing the accuracy between base and incremental classes. Although some existing FSCIL methods adopt prototype adjustment strategies (Zhang et al., 2021; Wang et al., 2023b), they typically calibrate some prototypes, especially incremental class prototypes, by incorporating aggregated information from other prototypes. In contrast, our method directly shifts base prototypes away from the distribution of incremental classes, which helps balance the refinement decision boundary and mitigate its bias toward base classes.

# 3 Methodology

## 3.1 Problem Formulation

In FSCIL, the model $f$ is trained on a sequence of datasets $\{\mathcal{D}_{train}^t\}_{t=0}^T$, where $\mathcal{D}_{train}^t = \{(\mathbf{x}_i, y_i)\}_i$ is the training set from session $t$ and $\mathbf{x}_i$ is a sample from class $y_i \in \mathcal{C}^t$. $\mathcal{C}^t$ is the label set of dataset $\mathcal{D}_{train}^t$. Usually,

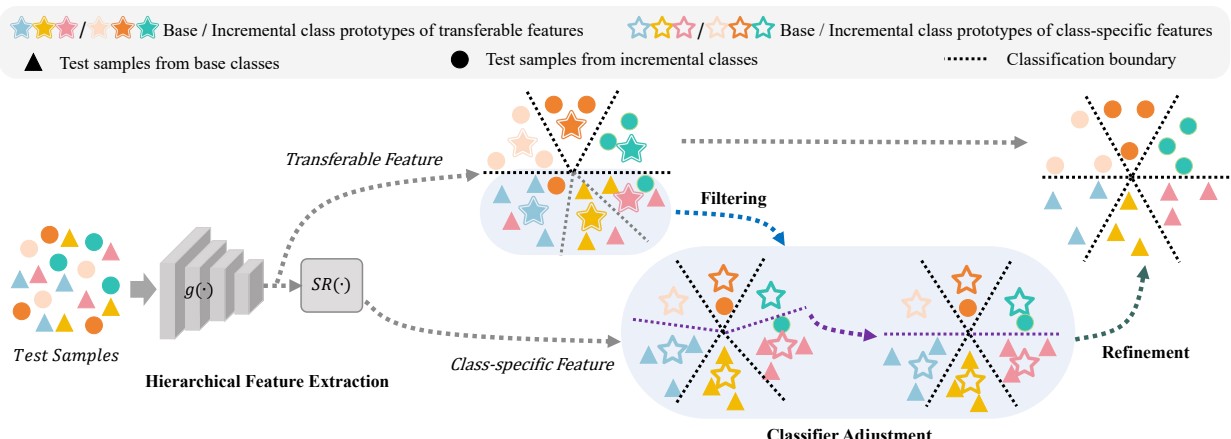

Figure 1: Illustration of the proposed hierarchical filtering and refinement classification. Test samples with different labels, where each color represents a distinct class. $g(\cdot)$ denotes the backbone feature extractor that produces transferable features, while $SR(\cdot)$ represents the Selection and Reorganization module that reorganizes these features to produce class-specific features.

the training set $\mathcal{D}_{train}^{0}$ in the base session contains sufficient samples, and the training set $\mathcal{D}_{train}^{t}(t \geq 1)$ with the limited samples in each incremental session can be organized as $N$-way $K$-shot format, i.e., there are only $K$ samples for each of the $N$ classes from $\mathcal{C}^{t}$. A model in each session $t$ can only access $\mathcal{D}_{train}^{t}$, but it needs to be tested on samples from all seen classes (i.e., $\mathcal{C}^{0} \cup \mathcal{C}^{1} \cdots \cup \mathcal{C}^{t}$). For clarity, the classes in $\mathcal{C}^{0}$ are defined as **base classes**, and the classes in $\mathcal{C}^{1} \cdots \cup \mathcal{C}^{t}$ are defined as **incremental classes**. The standard incremental learning paradigm strictly defines that each session has the same $N$ and $K$, and $\mathcal{C}^{i} \cap \mathcal{C}^{j} = \varnothing$ for $i \neq j$.

The training objective of the base session is to enable the model to learn to extract task-relevant and generalizable features and perform classification by minimizing a classification loss function. The model $f$ can be decomposed into linear classifiers $\eta$ and feature extractor $g$,

$$g(\mathbf{x}) = (\phi_1(\mathbf{x}), \phi_2(\mathbf{x}), \dots, \phi_D(\mathbf{x}))^\top, \quad \phi_l \in \Phi, \tag{1}$$

where $D$ is the vector dimension output by the feature extractor, $\phi_l$ is a feature mapping that maps $\mathbf{x}$ to $\mathbb{R}$ (including the process of global average pooling), so $\phi_l(\mathbf{x})$ can be referred to as a feature of $\mathbf{x}$. $g'(\mathbf{x}) = (\phi_1'(\mathbf{x}), \phi_2'(\mathbf{x}), \dots, \phi_D'(\mathbf{x}))^\top$ denotes the $L_2$-normalized values of $g(\mathbf{x})$, which is used in the subsequent definitions of transferable and class-specific features. A widely adopted baseline (Zhang et al., 2021) is to train model $f$ with classification loss in the base session, and then freeze $g$ in incremental sessions, thus significantly alleviating catastrophic forgetting and overfitting problems.

In addition, the mean vector of all training samples (i.e., the prototype) of each class $c$ is used as the classifier weight for this class in the incremental sessions, i.e., $h_c = \frac{1}{Num_c} \sum_{y_i=c} g(\mathbf{x}_i)$, where $Num_c$ indicates the number of training samples in class $c$. During inference, nearest neighbor prototype classifiers are used.

### 3.2 Decomposition of FSCIL

In the incremental session $t'$, for any test sample $\mathbf{x}$, the general objective of the FSCIL problem is to predict the probability $P(\mathbf{x} \in \mathcal{X}_c | \mathcal{D}_{train}^{t'}, f)$, where $\mathcal{X}_c$ denotes the domain of label $c$. However, due to the distinct roles of base and incremental classes in FSCIL, as discussed in the introduction, it is difficult to simultaneously meet their performance requirements under a unified classification objective.

Considering that the label sets of base and incremental classes are disjoint, this objective can be reformulated as:

$$P(\mathbf{x} \in \mathcal{X}_{i,j} | \mathcal{D}_{train}^{t'}, f), \tag{2}$$

where $i \in \{\text{base}, \text{incremental}\}$ indicates whether the class belongs to the base or incremental group, and $j$ denotes the index of a specific class within the corresponding group. Thus, the pair $(i, j)$ represents the label of $\mathbf{x}$. Then, using the chain rule of probability, the above probability can be decomposed as:

$$\underbrace{P(\mathbf{x} \in \mathcal{X}_i \mid \mathcal{D}_{\text{train}}^{t'}, f)}_{\text{(1) Inter-group classification}} \cdot \underbrace{P(\mathbf{x} \in \mathcal{X}_{i,j} \mid \mathbf{x} \in \mathcal{X}_i, \mathcal{D}_{\text{train}}^{t'}, f)}_{\text{(2) Intra-group classification}} \tag{3}$$

- (1) determines whether $\mathbf{x}$ belongs to the base or incremental group, i.e., determining $i$.

- (2) identifies which specific class $j$ the sample belongs to within the predicted group $i$.

Let $\hat{i}$ and $\hat{j}$ be the ground truth of $\mathbf{x}$. Then, Equation 3 indicates that optimizing overall classification objective can be decomposed into improving $P(\mathbf{x} \in \mathcal{X}_{\hat{i}} \mid \mathcal{D}_{\text{train}}^{t'}, f)$ and $P(\mathbf{x} \in \mathcal{X}_{\hat{i},\hat{j}} \mid \mathbf{x} \in \mathcal{X}_{\hat{i}}, \mathcal{D}_{\text{train}}^{t'}, f)$, which makes it easier to simultaneously accommodate the performance requirements of base and incremental classes.

Motivated by the above probabilistic decomposition, we propose a novel method called **Hierarchical Filtering and Refinement Classification (HFRC)** to address the above sub-objectives, as illustrated in Figure 1. During the base training stage, we employ hierarchical feature extraction (Section 3.3) to separately obtain two types of features that are beneficial for classifying base and incremental classes, respectively. These features are then exploited to perform filtering and refinement classification (Section 3.4), aiming to achieve accurate classification within incremental classes, between base and incremental classes, and within base classes.

## 3.3 Hierarchical Feature Extraction

To enable balanced accuracy for both base and incremental classes, the feature extractor trained during the base sessions should strive to obtain features that are descriptive for both the currently visible base classes and the unseen future incremental classes. However, the feature extractor directly involved in base training inevitably tends to focus on class-specific features for base classes, while losing transferable features may be useful for incremental classes but interfere with current training process.

Let $\phi_l$ be a feature mapping that maps the input $\mathbf{x}_i$ into a feature $\phi_l(\mathbf{x}_i)$, we formalize the conditions for the two kinds of features for a set of classes $\mathcal{C}^b$ as follows:

- *Class-specific Features*: For most $\mathbf{x}_i$ $(y_i \in \mathcal{C}^b)$, $\phi_l'(\mathbf{x}_i)$ is a distinct value close to 0 or 1, and there exists at least one $\mathbf{x}_i$ $(y_i \in \mathcal{C}^b)$ such that $\phi_l'(\mathbf{x}_i)$ approaches 1; For most $\mathbf{x}_i$ $(y_i \notin \mathcal{C}^b)$, $\phi_l'(\mathbf{x}_i)$ is a random value between 0 and 1 in the chaotic state.

- *Transferable Features*: For any $\mathbf{x}_i$, $\phi_l'(\mathbf{x}_i)$ is a relatively distinct value between 0 and 1.

To separately obtain the two types of features described above, we introduce an isolating module that decouples target features and the classification task, thereby enabling hierarchical feature extraction—transferable features are learned before the module, and class-specific features are derived after it. Specifically, we insert a Selection and Reorganization (SR) module after the original feature extractor to construct an enhanced feature extractor,

$$\tilde{g}(\mathbf{x}) = SR(g(\mathbf{x})) \in \mathbb{R}^{d'}, \tag{4}$$

where $SR$ denotes a module that consists of two fully connected layers and a ReLU activation function. Guided by the loss function during training, $\tilde{g}(\mathbf{x})$, i.e., class-specific features, becomes more discriminative for recognizing base training classes through the process of selecting and reorganization. Meanwhile, $g$ is encouraged to extract richer $g(\mathbf{x})$, i.e., transferable features, for subsequent selection and reorganization.

**Gradient Analysis**  To better understand how the SR module causally encourages transferable feature learning, we analyze the gradient behavior from the perspective of the optimization objective.

During base training, the classification objective can be written as $\mathcal{L}\big(\{\mathcal{S}(\eta_c, \tilde{g}(\mathbf{x}))\}_{c \in \mathcal{C}_0}, y\big)$, where $\mathcal{S}(\cdot, \cdot)$ denotes the cosine similarity function.

For $\tilde{g}(x)$, the gradient direction is strongly determined by the classifier parameters $\eta_c$, i.e.,

$$\frac{\partial \mathcal{L}}{\partial \tilde{g}(\mathbf{x})} = \sum_{c \in \mathcal{C}_0} \frac{\partial \mathcal{L}}{\partial \mathcal{S}(\eta_c, \tilde{g}(\mathbf{x}))} \frac{\partial \mathcal{S}(\eta_c, \tilde{g}(\mathbf{x}))}{\partial \tilde{g}(\mathbf{x})} \in \text{span}\{\eta_c \mid c \in \mathcal{C}_0\}. \tag{5}$$

Since the number of base classes is limited, the subspace spanned by $\{\eta_c\}$ typically has much lower dimensionality than the full feature space. Consequently, there exists a subset of feature dimensions $D_{\text{base}} \subset \{1, \ldots, D\}$ with $|D_{\text{base}}| \ll D$ such that

$$\left| \frac{\partial \mathcal{L}}{\partial \tilde{g}_d(x)} \right| \approx 0, \quad \forall d \notin D_{\text{base}}, \tag{6}$$

implying that only a small number of feature channels are strongly activated by base class supervision.

For the backbone feature $g(\mathbf{x})$, applying the chain rule yields

$$\frac{\partial \mathcal{L}}{\partial g(\mathbf{x})} = \frac{\partial \tilde{g}(\mathbf{x})}{\partial g(\mathbf{x})} \frac{\partial \mathcal{L}}{\partial \tilde{g}(\mathbf{x})}, \tag{7}$$

where $\frac{\partial \tilde{g}(\mathbf{x})}{\partial g(\mathbf{x})}$ denotes the Jacobian of the SR module in the backward pass. From a forward perspective, the SR parameters are directly optimized to absorb the discriminative gradients required for base class classification. From a backward perspective, the SR module acts as a gradient reallocation mechanism, redistributing the gradients propagated back to the backbone through its Jacobian. As a result, excessive optimization pressure toward a small set of class-specific directions is alleviated, encouraging the backbone to preserve more transferable feature representations. A corresponding visualization analysis is provided in Section 4.4.1.

The subsequent sections consistently employ '$\sim$' to distinguish symbols associated with the two types of features.

### 3.4 Filtering and Refinement Classification

With the two types of features obtained, we are able to leverage their respective strengths through a hierarchical classification process—including filtering and refinement—to comprehensively address the classification sub-objectives derived in Section 3.2.

For the base session, classification is directly conducted using class-specific features $\tilde{g}(\mathbf{x})$ and their corresponding prototype classifiers $\tilde{h}_c$, where the predicted label is determined by the maximum cosine similarity. This enables higher accuracy on base classes due to the enhanced discriminability of $\tilde{g}(\mathbf{x})$.

During the incremental sessions, prototype classifiers $h$ and $\tilde{h}$ are respectively constructed from the transferable and class-specific features of the training samples. Given test image $\mathbf{x}$, its transferable feature $g(\mathbf{x})$ and class-specific feature $\tilde{g}(\mathbf{x})$ are obtained separately.

Firstly, transferable features $g(\mathbf{x})$ and their corresponding prototype classifiers $h_c$ are employed to perform a preliminary classification,

$$y^\star = \underset{c \in \cup_{t=0}^{t'} \mathcal{C}^t}{\text{argmax}} \, \mathcal{S}(g(\mathbf{x}), h_c), \tag{8}$$

On the one hand, the transferable features enable classification within incremental classes. On the other hand, they facilitate a coarse separation between base and incremental classes, allowing us to filter out samples initially identified as belonging to base classes. Thereafter, the samples $\mathbf{x}$ that are filtered out as base candidates will undergo refinement classification using their class-specific features $\tilde{g}(\mathbf{x})$ and the corresponding prototype classifiers $\tilde{h}$,

$$y^\star = \underset{c \in \cup_{t=0}^{t'} \mathcal{C}^t}{\text{argmax}} \, \mathcal{S}(\tilde{g}(\mathbf{x}), \tilde{h}_c). \tag{9}$$

Leveraging class-specific features not only enables more accurate classification within base classes but also allows for finer-grained distinction between true base class samples and incremental samples that are similar to them.

---

**Algorithm 1** The overall procedure of the incremental session.

---

**Input:** Training set $\mathcal{D}_{train}^{t'}$ in session $t'(t' > 0)$, test sample $\mathbf{x}$, feature extractor $\tilde{g}$ (including $g$ and $SR$),
    classifiers $h$ and $\tilde{h}$ for old classes, and $\Delta \in \mathbb{R}^{|\mathcal{C}^0|}$.

**Output:** Classification result $y^\star$ for $\mathbf{x}$.

1: $\mathcal{F}, \tilde{\mathcal{F}} \leftarrow$ Extract transferable features and class-specific features from $\mathcal{D}_{train}^t$ using $g$ and $\tilde{g}$ (Equation 4)
2: $h_{novel}, \tilde{h}_{novel} \leftarrow$ Generate novel prototype classifiers from $\mathcal{F}, \tilde{\mathcal{F}}$
3: Update $\Delta$ based on $\tilde{h}_{novel}$ (Equation 11)
4: $h, \tilde{h} \leftarrow h \cup h_{novel}, \tilde{h} \cup \tilde{h}_{novel}$ // Update classifiers
5: Calculate $\{\tilde{h}_c'|c \in \mathcal{C}^0\}$ based on $\Delta$ and original base prototypes $\{\tilde{h}|c \in \mathcal{C}^0\}$ (Equation 10)
6: $y^\star \leftarrow$ Classify $g(\mathbf{x})$ using classifiers $h$ (Equation 8)
7: **if** $y^\star$ in $\mathcal{C}^0$ **then**
8:     $y^\star \leftarrow$ Classify $\tilde{g}(\mathbf{x})$ using classifiers $\{\tilde{h}_c'|c \in \mathcal{C}^0\}$ and $\{\tilde{h}_c|c \in \cup_{t=1}^{t'}\mathcal{C}^t\}$ (Equation 9)
9: **end if**

---

**Classifier Adjustment** As previously mentioned, the classifiers $\tilde{h}$ serve to distinguish between base and incremental classes that are easily confused. Thus, to better balance the classification accuracy between base and incremental classes, we apply a targeted adjustment of base class prototypes $\tilde{h}$ based on the distribution of incremental classes. Considering that continuous optimization may reintroduce the issue of catastrophic forgetting for base classes, this adjustment is designed as a one-time process.

Specifically, $\tilde{h}_c(c \in \mathcal{C}^0)$ are calculated as described below to obtain the adjusted prototypes $\tilde{h}_c'$ for inference, while the original prototypes $\tilde{h}_c$ remains unchanged,

$$\tilde{h}_c' = \tilde{h}_c - \gamma \cdot \frac{\Delta_c}{\|\Delta_c\|}, c \in \mathcal{C}^0, \tag{10}$$

where $\gamma$ is a hyperparameter used to amplify the adjustment. $\Delta_c$ summarizes the main locations where incremental classes exist for $\tilde{h}_c$, and is continuously updated when novel incremental classes arrive,

$$\Delta_c \leftarrow \Delta_c + \sum_{i \in \mathcal{C}^t} \max\left(\mathcal{S}(\tilde{h}_c, \tilde{h}_i), 0\right) \cdot \frac{\tilde{h}_i}{\|\tilde{h}_i\|}, \tag{11}$$

where $\max(x, 0)$ is used to select the novel class prototype $\tilde{h}_i$ that has a cosine similarity greater than 0 with the base class prototype $\tilde{h}_c$, and $\Delta_c$ is initialized as $\mathbf{0}$.

After integrating the classifier adjustment, the overall procedure of the incremental session is detailed in Algorithm 1.

# 4 Experiments

## 4.1 Experimental Setup

### 4.1.1 Datasets

Following Tao et al. (2020), we conduct experiments on three datasets: *mini*ImageNet (Russakovsky et al., 2015), CIFAR100 (Krizhevsky, 2009), and CUB200 (Wah et al., 2011). *mini*ImageNet is a subset of the ImageNet dataset, comprising 600 images per class, with 500 allocated for training and 100 for testing purposes. Similarly, each class of CIFAR100 consists of 500 training images and 100 testing images. CUB200 is a fine-grained dataset that includes 11,788 images across 200 classes. The statistic characteristics of three datasets are listed in Table 1.

Table 1: Statistics of datasets. $\left|\mathcal{C}^0\right|$: number of base classes. $T$: number of incremental sessions.

| Dataset | $\left|\mathcal{C}^0\right|$ | $T$ | $N$ | $K$ | Resolution |
|---|---|---|---|---|---|
| *mini*ImageNet | 60 | 8 | 5 | 5 | 84×84 |
| CIFAR100 | 60 | 8 | 5 | 5 | 32×32 |
| CUB200 | 100 | 10 | 10 | 5 | 224×224 |

Table 2: Comparison with the state-of-the-art methods on *mini*ImageNet dataset. **Overall acc.** and **Inc. acc.** denote overall accuracy and incremental class accuracy, respectively.

| Method | Overall acc. in each session (%) | | | | | | | | | Inc. acc. |
|---|---|---|---|---|---|---|---|---|---|---|
| | 0 | 1 | 2 | 3 | 4 | 5 | 6 | 7 | 8 | Avg |
| TOPIC (Tao et al., 2020) | 61.31 | 50.09 | 45.17 | 41.16 | 37.48 | 35.52 | 32.19 | 29.46 | 24.42 | - |
| CEC (Zhang et al., 2021) | 72.00 | 66.83 | 62.97 | 59.43 | 56.70 | 53.73 | 51.19 | 49.24 | 47.63 | 15.27 |
| FACT (Zhou et al., 2022) | 75.32 | 70.34 | 65.84 | 62.05 | 58.68 | 55.35 | 52.42 | 50.42 | 48.51 | 13.98 |
| C-FSCIL (Hersche et al., 2022) | 76.40 | 71.14 | 66.46 | 63.29 | 60.42 | 57.46 | 54.78 | 53.11 | 51.41 | 19.08 |
| CLOM (Zou et al., 2022) | 72.08 | 67.28 | 63.30 | 59.85 | 56.82 | 53.82 | 51.08 | 49.15 | 47.95 | 13.58 |
| LIMIT (Zhou et al., 2023) | 72.32 | 68.47 | 64.30 | 60.78 | 57.95 | 55.07 | 52.70 | 50.72 | 49.19 | 21.53 |
| Bidist (Zhao et al., 2023) | 74.65 | 70.43 | 66.29 | 62.77 | 60.75 | 57.24 | 54.79 | 53.65 | 52.22 | 27.73 |
| TEEN (Wang et al., 2023b) | 73.53 | 70.55 | 66.37 | 63.23 | 60.53 | 57.95 | 55.24 | 53.44 | 52.08 | 32.19 |
| SAVC (Song et al., 2023) | 81.12 | 76.14 | 72.43 | 68.92 | 66.48 | 62.95 | 59.92 | 58.39 | 57.11 | 28.60 |
| OSHHG (Cui et al., 2024) | 60.65 | 59.00 | 56.59 | 54.78 | 53.02 | 50.73 | 48.46 | 47.34 | 46.75 | 29.30 |
| EHS (Deng & Xiang, 2024) | 71.25 | 66.65 | 62.84 | 59.65 | 56.90 | 54.14 | 51.63 | 50.05 | 49.06 | - |
| M2SD (Lin et al., 2024) | 82.11 | **79.92** | 75.44 | 71.31 | 68.29 | 64.32 | 61.13 | 58.64 | 56.51 | - |
| MICS (Kim et al., 2024) | 84.40 | 79.48 | 75.09 | 71.40 | 68.89 | 66.16 | 63.57 | 61.79 | 60.74 | 34.07 |
| DyCR (Pan et al., 2025) | 73.18 | 70.16 | 66.87 | 63.43 | 61.18 | 58.79 | 55.00 | 52.87 | 51.08 | 13.08 |
| CSR (Hu et al., 2025b) | 80.90 | 75.89 | 71.80 | 68.59 | 65.86 | 62.41 | 59.33 | 57.71 | 56.41 | 31.23 |
| D2A (Zhao et al., 2025) | 81.20 | 75.06 | 71.44 | 68.09 | 66.16 | 63.05 | 60.02 | 58.48 | 57.72 | 38.14 |
| PGLS (Hu et al., 2025a) | 84.40 | 79.52 | 75.05 | 71.22 | 67.87 | 64.92 | 61.82 | 59.87 | 58.57 | - |
| Ours | **86.12** | 79.74 | **75.64** | **72.24** | **69.51** | **66.66** | **63.94** | **62.17** | **61.15** | **44.53** |

### 4.1.2 Implementation Details

Following Tao et al. (2020), we employ ResNet18 (He et al., 2016) as the backbone, and the network for CUB200 is initialized by ImageNet (Deng et al., 2009) pre-trained parameters. The results of the comparative methods that are not reported in their papers are reproduced by their publicly available source code. Since the feature extractor obtained from the base session can be directly utilized in the incremental sessions, our method only requires a simple classifier computation based on Equation 10 and Equation 11, making it more efficient than FSCIL methods that rely on backpropagation-based optimization. The additional overhead introduced during inference is minimal and can be considered negligible. We train the model for 50 epochs on CUB200 and 100 epochs on *mini*ImageNet and CIFAR100. The initial learning rate is set to 0.01 for *mini*ImageNet and CIFAR100 datasets, and 0.001 for CUB200 dataset. The output feature size of two fully connected layers of the SR module is 2048. In Equation 10, $\gamma$ is set to 7 for all datasets. More details about the experimental setup are included in the appendix.

### 4.2 Main Results

We compare our method with recent state-of-the-arts FSCIL methods on three widely used datasets. Following D2A, we report both overall accuracy and incremental class accuracy to better illustrate the method's

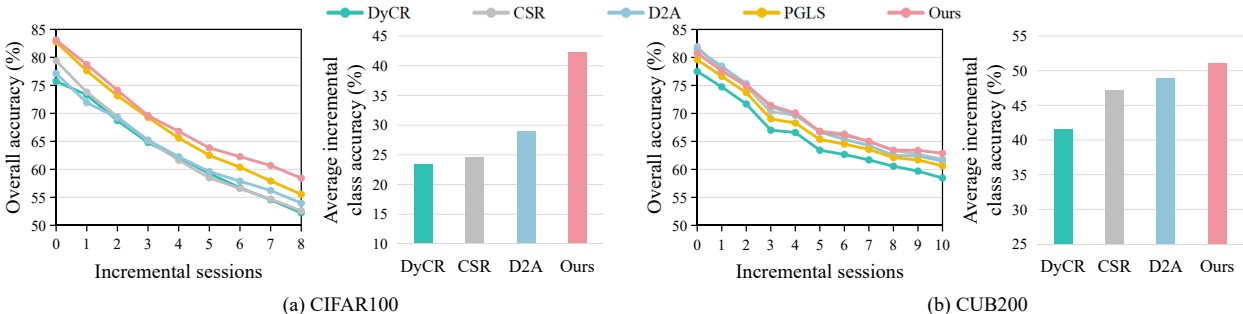

Figure 2: Comparison with the state-of-the-art methods on CIFAR100 and CUB200 datasets.

Table 3: Ablation studies of our proposed method on CIFAR100 dataset.

| HF | FC | RC | CA | **Overall acc.** in each session (%) | | | | | | | | | Inc. acc. |
|----|----|----|----|-------|-------|-------|-------|-------|-------|-------|-------|-------|-----------|
| | | | | 0 | 1 | 2 | 3 | 4 | 5 | 6 | 7 | 8 | Avg |
| | | | | 82.78 | 78.05 | 73.37 | 68.49 | 64.90 | 61.48 | 58.47 | 55.84 | 53.23 | 23.73 |
| √ | | | | **83.10** | 77.82 | 72.57 | 68.24 | 64.61 | 61.78 | 58.78 | 56.18 | 53.75 | 20.40 |
| √ | √ | | | **83.10** | 76.20 | 72.99 | 69.01 | 66.93 | 64.12 | 62.90 | 61.15 | 58.89 | 35.06 |
| √ | √ | √ | | **83.10** | **79.40** | **75.21** | 70.95 | **68.14** | **65.04** | **63.50** | **61.79** | **59.48** | 40.18 |
| √ | √ | √ | √ | **83.10** | 78.72 | 74.13 | 69.60 | 66.79 | 63.84 | 62.26 | 60.68 | 58.62 | **43.21** |

capability in balancing performance between base and incremental classes. The results are shown in Table 2 and Figure 2.

It can be observed that our method achieves the best overall performance, with particularly significant improvements in incremental class accuracy. Specifically, compared to recent methods such as CSR and D2A, which also focus on improving incremental class performance, our method still shows clear advantages in incremental class accuracy, while also maintaining superior overall accuracy. Besides, we leverage class-specific features to achieve better classification performance within base classes, resulting in improved accuracy in the base session. For subsequent incremental sessions, we fully utilize the strengths of both types of features to perform classification, enabling our method to sustain high overall accuracy even as more incremental class samples are introduced. Although our method performs comparably to M2SD in the first incremental session, by the final incremental session, it surpasses M2SD by 4.64%. This demonstrates that our method, by comprehensively addressing the classification sub-objectives, is more effective at improving FSCIL performance. More experimental results are presented in the appendix.

### 4.3 Ablation Study

To analyze the role of different components in our method, we conduct ablation studies in Table 3.

Firstly, hierarchical feature extraction (**HF**) under the standard classification strategy, as described in Section 3.3, improves base class accuracy by performing selection and reorganization of features for base classes. However, the refined features tailored for base classes do not generalize well to incremental classes, leading to a decline in incremental class accuracy.

On this basis, during the incremental sessions, applying only the filtering (**FC**) in the filtering and refinement classification as described in Equation 8 enables more accurate classification within incremental classes by leveraging transferable features. This also facilitates a coarse separation between base and incremental classes, thereby enhancing the overall accuracy in subsequent sessions. Further applying refinement (**RC**) as described in Equation 9 introduces class-specific features, allowing for finer-grained distinctions between base classes and incremental classes that are similar to them. This also enables more accurate discrimination

among base classes, resulting in additional performance gains. Finally, classifier adjustment (**CA**) as defined in Equation 10 and Equation 11 further balances the accuracy between base and incremental classes, thereby improving incremental class accuracy. Although a slight decrease in overall accuracy is observed due to the high proportion of base class samples during evaluation, our method still achieves superior overall accuracy compared to existing approaches (Figure 2a).

### 4.4 Further Analysis

#### 4.4.1 Visualization of Feature Vectors

We visualize the $L_2$ normalized feature vectors on the CIFAR100 test set in Figure 3, which demonstrates that hierarchical feature extraction (**HF**) can indeed stimulate $g$ to learn and retain transferable features for incremental classes.

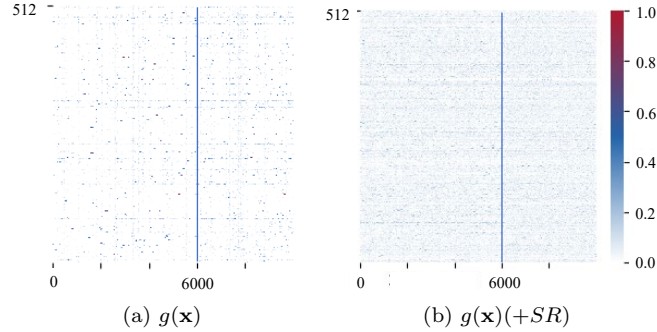

(a) $g(\mathbf{x})$   (b) $g(\mathbf{x})(+SR)$

Figure 3: Comparison of heatmaps of $L_2$ normalized feature vectors on whether adding $SR$ after $g$. The higher the value, the stronger the discriminative power of features.

Specifically, more features are activated in Figure 3b, indicating that adding $SR$ after $g$ indeed stimulates $g$ to learn and retain more features. Moreover, there is no obvious difference in the activation density and intensity of pixels between the first 6000 base class samples and the last 4000 incremental class samples. This indicates that these features are transferable features, suggesting that feature extraction is basically not biased towards base classes. In Figure 3a, the activation density of pixels for base classes is notably lower, with generally higher or lower activation values. This implies that the few feature mappings utilized for base classes exhibit a superiority in discriminative ability compared to other feature mappings. Other feature mappings deemed to interfere with base training classes, yet applicable for recognizing future incremental classes, have been weakened or abandoned throughout the base training process. Consequently, we preserve transferable features before $SR$ to complement the final class-specific features.

#### 4.4.2 Visualization of Feature Space

We visualize the feature space of feature vectors $g(\mathbf{x})$ on the CIFAR100 test set with t-SNE (van der Maaten & Hinton, 2008) in Figure 4. To make the results clearer, we randomly select five incremental classes. It can be observed that incremental class samples in the baseline are mostly mapped to the base class positions and cannot cluster well due to the lack of effective discriminative features (see Figure 4a). In Figure 4b, transferable feature $g(\mathbf{x})$ obtained by hierarchical feature extraction (**HF**) (i.e., adding the SR module after $g$) further makes incremental classes form effective clusters based on semantic categories in the feature space.

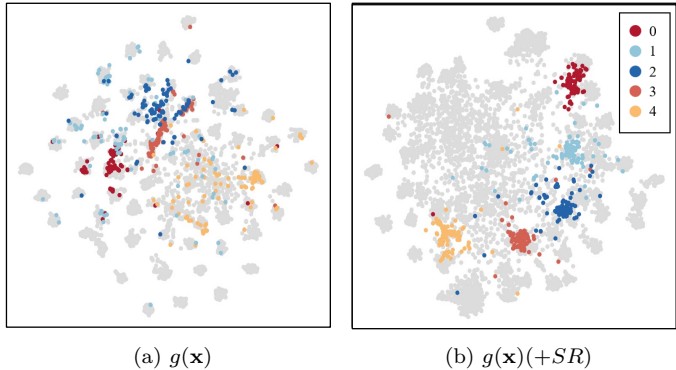

(a) $g(\mathbf{x})$   (b) $g(\mathbf{x})(+SR)$

Figure 4: Visualization of feature space with t-SNE. Gray represents base classes and other colors represent incremental classes.

However, it also blurs the boundaries among base classes and between base classes and incremental classes. Therefore, we propose filtering and refinement classification to intelligently combine transferable features $g(\mathbf{x})$ with class-specific features $\tilde{g}(\mathbf{x})$.

Table 4: Comparison with different classification strategies on CIFAR100 dataset.

| Strategy | Overall acc. in each session (%) | | | | | | | | | Inc. acc. |
| | 0 | 1 | 2 | 3 | 4 | 5 | 6 | 7 | 8 | Avg |
|---|---|---|---|---|---|---|---|---|---|---|
| **Pre** | 82.92 | 77.91 | 72.63 | 68.32 | 64.76 | 61.89 | 59.03 | 56.42 | 54.07 | 20.92 |
| **Post** | 78.82 | 74.38 | 70.19 | 67.03 | 63.99 | 60.91 | 58.63 | 56.43 | 54.77 | 18.99 |
| **AD** | **83.10** | 78.46 | 73.64 | 69.36 | 66.35 | 63.49 | 61.39 | 59.34 | 57.02 | 19.63 |
| Ours | **83.10** | **79.40** | **75.21** | **70.95** | **68.14** | **65.04** | **63.50** | **61.79** | **59.48** | **40.18** |

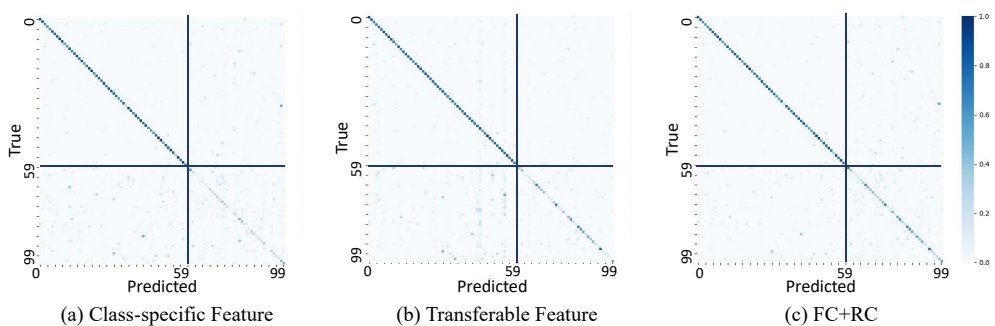

(a) Class-specific Feature  (b) Transferable Feature  (c) FC+RC

Figure 5: Confusion matrices of classification results in the last session on CIFAR100 dataset.

### 4.4.3 Analyses of Classification Strategies

We compare the performance of different classification strategies in Table 4. No matter whether it is pre-integration (**Pre**, i.e., feature vector integration), post-integration (**Post**, i.e., similarity integration), or the idea of anomaly detection (**AD**, i.e., first use class-specific features to detect samples that do not belong to base classes, and then use their transferable features to reclassify them), they all get a low incremental class accuracy. This indicates that even if two types of features are extracted, it remains very challenging to fully leverage their respective advantages simultaneously. Our filtering and refinement classification can achieve the highest incremental class accuracy, while boosting overall accuracy.

To further provide an intuitive understanding of how the proposed decomposition resolves the base–incremental tension, Figure 5 visualizes the confusion matrices under different feature representations, where classes 0–59 correspond to base classes and 60–99 correspond to incremental classes.

As shown in Figure 5a, conventional methods that directly optimize the classification objective using class-specific features achieve high accuracy on base classes, but suffer from severe misclassification on incremental classes. In contrast, when using transferable features, several important observations can be made from Figure 5b: (1) Almost no base class samples are misclassified as incremental classes (the upper-right block), indicating low mis-filtering; (2) Confusion among incremental classes (the lower-right block) is significantly reduced compared to class-specific features; (3) Errors caused by incremental samples being misclassified as base classes (the lower-left block) are noticeably more frequent than confusions among incremental classes themselves. These observations indicate that, when using transferable features, most errors originate from samples being routed into the base group. Therefore, we apply class-specific features only to samples filtered into the base group for refinement. As shown in Figure 5c, after introducing the refinement stage, misclassifications (particularly those in the lower-left block) are substantially reduced, leading to a further increase in correctly classified samples. The ablation results in Table 3 corroborate these observations, where both the filtering (**FC**) and the refinement (**RC**) consistently improve classification accuracy.

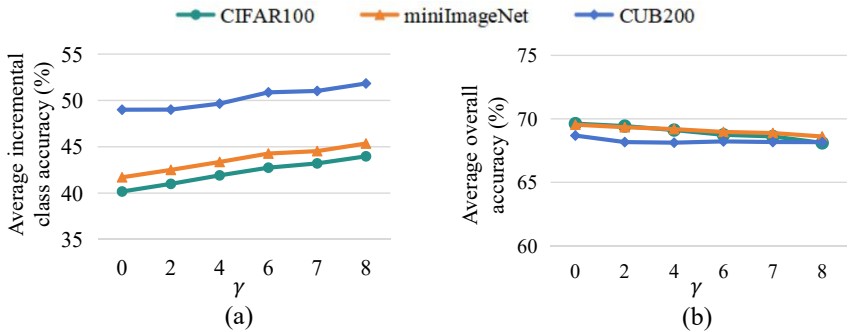

Figure 6: The impact of the hyperparameter $\gamma$.

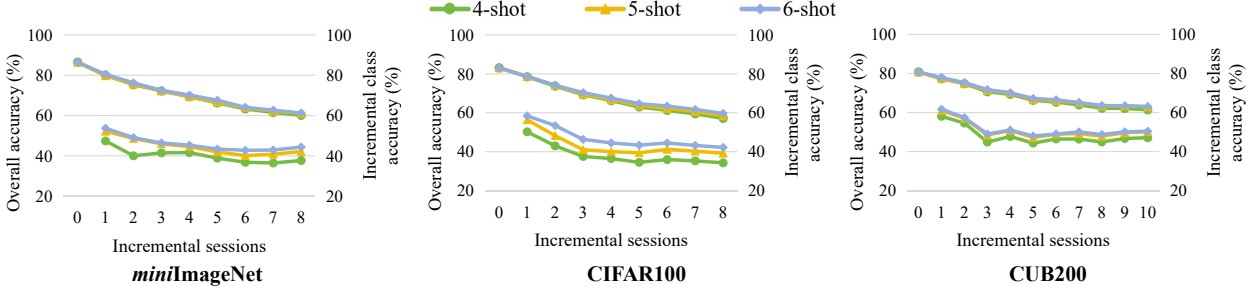

Figure 7: Performance under varying numbers of shots across datasets.

### 4.4.4 Hyperparameter Analyses

We introduce the hyperparameter $\gamma$ in the classifier adjustment (**CA**) module to control the degree of adjustment applied to base class classifiers. To analyze its impact on accuracy, we report results for different $\gamma$ values, showing consistent performance trends across all datasets. As shown in Figure 6, applying **CA** (i.e., $\gamma > 0$) moves the base class classifiers farther from the incremental class classifiers, which leads to a significant improvement in incremental class accuracy (Figure 6a), thereby further balancing the accuracy between base and incremental classes. At the same time, as the hyperparameter $\gamma$ increases, overall accuracy continues to slightly decline, implying that $\gamma$ can control the trade-off between overall accuracy and accuracy balance. Consequently, we set $\gamma = 7$ in our experiments, as it maintains overall accuracy that outperforms other methods while achieving as much accuracy balance as possible.

### 4.4.5 Broader Sensitivity Analyses

To further evaluate robustness, we conduct experiments under several protocol variations, including different numbers of shots and ways, different base class splits, and multiple random seeds across datasets.

As shown in Figures 7 and 8, changing the number of shots or ways exhibits stable and consistent trends across all datasets. As expected, performance steadily improves as the number of shots increases. When varying the number of ways, the performance remains similar for sessions with the same number of seen classes, indicating that our method is insensitive to the specific way partitioning. As reported in Table 5, we evaluate our method under three randomly sampled base class splits and three different random seeds. Compared with the standard setting, the results exhibit only minor variations, demonstrating stable performance. Overall, these experiments demonstrate the robustness of our method across different datasets and protocol variations.

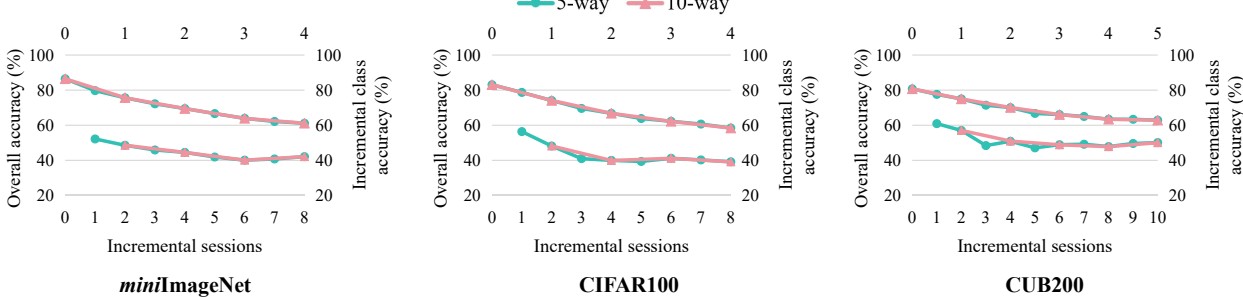

Figure 8: Performance under varying numbers of ways across datasets.

Table 5: Average accuracy with ± standard deviation under different base splits and random seeds across datasets.

| | *mini*ImageNet | | CIFAR100 | | CUB200 | |
|---|---|---|---|---|---|---|
| | Overall acc. | Inc. acc. | Overall acc. | Inc. acc. | Overall acc. | Inc. acc. |
| Base splits | 70.21±0.88 | 43.71±1.16 | 68.25±0.52 | 42.61±0.85 | 69.79±0.67 | 51.69±0.94 |
| Random seeds | 70.43±0.57 | 43.95±0.82 | 68.74±0.18 | 42.99±0.31 | 69.09±0.32 | 50.50±0.74 |

Table 6: Runtime and memory comparison with recent methods.

| Method | Training time (s) | Inference time (ms) | Memory (MB) |
|---|---|---|---|
| SAVC (Song et al., 2023) | 22.09 | 1.06 | 299.68 |
| CSR (Hu et al., 2025b) | 43.37 | 1.09 | 306.05 |
| D2A (Zhao et al., 2025) | 22.05 | 1.27 | 300.18 |
| Ours | 2.94 | 1.08 | 280.15 |

### 4.4.6 Runtime and Memory Analyses

To provide a clearer evaluation of incremental efficiency, we compare training time, inference time, and memory consumption with recent and state-of-the-art methods on the CUB200 dataset in Table 6. The results are consistent with the algorithmic analysis presented in Section 4.1.2 of the paper. During the incremental session, the computation only involves a simple classifier update, without any backpropagation-based optimization. As a result, our method significantly reduces the training time in the incremental session compared with recent approaches. Furthermore, during inference, our method only introduces a lightweight forward computation and similarity calculation. Therefore, it has negligible impact on inference time, and is even slightly faster than methods that also introduce additional strategy during inference (e.g., D2A). Moreover, our method requires a comparable amount of memory to other methods during training.

## 5   Conclusion

In this paper, considering that base and incremental classes play different yet both critical roles in few-shot class-incremental learning (FSCIL), we approach FSCIL from a more structured perspective by decomposing the overall classification objective into three sub-objectives: classification between base and incremental classes, and intra-group classification within each of them. Building on this insight, we propose a novel classification framework, termed Hierarchical Filtering and Refinement Classification (HFRC), which hierarchically addresses these decomposed classification sub-objectives. Extensive experiments on multiple benchmark datasets demonstrate that our method not only effectively balances the classification accuracy between base and incremental classes, but also achieves state-of-the-art performance, highlighting its potential as a comprehensive and practical solution to the challenges inherent in FSCIL.

**Acknowledgments**

This work was supported in part by the National Natural Science Foundation of China under Grant 62202272, 62172256, 62202278, in part by the Natural Science Foundation of Shandong Province under Grant ZR2024LZH002 and Taishan Scholar Project of Shandong Province under Grant tstp20250704.

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

## A   Detailed Experimental Setup

Our method is conducted with PyTorch library on a single NVIDIA A800 and SGD with momentum is used for optimization. Our code is based on the code released by CEC (Zhang et al., 2021) under the MIT license. We adopt the standard data preprocessing, including random resizing, random horizontal flipping, and color jittering in the training process. Moreover, we adopt a data augmentation strategy similar to that used in SAVC (Song et al., 2023), which helps achieve higher accuracy in the base session. Notably, our method can significantly improve the accuracy of incremental classes even under such high initial accuracy conditions, thereby achieving a better balance between base and incremental class performance.

## B   More Results

In this section, we report the overall accuracy and incremental class accuracy of each session on three datasets in Tables 7, 8, 9, 10, and 11.

Table 7: Comparison with the state-of-the-art methods on *mini*ImageNet dataset.

| Method | **Inc acc.** in each session (%) | | | | | | | |
|---|---|---|---|---|---|---|---|---|
| | 1 | 2 | 3 | 4 | 5 | 6 | 7 | 8 |
| CEC (Zhang et al., 2021) | 15.40 | 17.10 | 16.67 | 15.90 | 14.20 | 13.63 | 14.20 | 14.78 |
| FACT (Zhou et al., 2022) | 15.80 | 14.40 | 15.40 | 14.55 | 13.64 | 12.20 | 12.66 | 13.20 |
| C-FSCIL (Hersche et al., 2022) | 5.20 | 12.40 | 17.27 | 19.90 | 23.40 | 22.60 | 25.91 | 25.95 |
| CLOM (Zou et al., 2022) | 13.00 | 14.20 | 14.07 | 14.00 | 12.88 | 12.20 | 13.26 | 15.05 |
| LIMIT (Zhou et al., 2023) | 27.80 | 22.20 | 21.93 | 21.55 | 19.56 | 19.53 | 19.54 | 20.10 |
| Bidist (Zhao et al., 2023) | 27.00 | 30.30 | 29.60 | 27.85 | 28.44 | 26.80 | 26.23 | 25.62 |
| TEEN (Wang et al., 2023b) | 40.20 | 35.60 | 32.47 | 32.70 | 29.96 | 28.33 | 28.89 | 29.35 |
| SAVC (Song et al., 2023) | 33.80 | 30.30 | 29.67 | 30.50 | 27.32 | 25.20 | 25.46 | 26.58 |
| OSHHG (Cui et al., 2024) | 39.26 | 32.26 | 31.31 | 30.12 | 26.95 | 24.08 | 24.51 | 25.89 |
| MICS (Kim et al., 2024) | 43.00 | 34.50 | 31.73 | 33.65 | 32.48 | 31.34 | 32.14 | 33.73 |
| DyCR (Pan et al., 2025) | 14.40 | 13.80 | 12.93 | 13.95 | 12.12 | 11.20 | 12.31 | 13.90 |
| CSR (Hu et al., 2025b) | 37.60 | 36.20 | 32.73 | 32.70 | 28.36 | 26.43 | 27.26 | 28.55 |
| D2A (Zhao et al., 2025) | 46.00 | 40.20 | 38.40 | 40.05 | 36.12 | 33.83 | 34.17 | 36.33 |
| Ours | **52.20** | **48.60** | **45.93** | **44.55** | **41.92** | **40.13** | **40.80** | **42.13** |

Table 8: Comparison with the state-of-the-art methods on CIFAR100 dataset.

| Method | Overall acc. in each session (%) | | | | | | | | |
|---|---|---|---|---|---|---|---|---|---|
| | 0 | 1 | 2 | 3 | 4 | 5 | 6 | 7 | 8 |
| TOPIC (Tao et al., 2020) | 64.10 | 55.88 | 47.07 | 45.16 | 40.11 | 36.38 | 33.96 | 31.55 | 29.37 |
| CEC (Zhang et al., 2021) | 73.07 | 68.88 | 65.26 | 61.19 | 58.09 | 55.57 | 53.22 | 51.34 | 49.14 |
| FACT (Zhou et al., 2022) | 78.80 | 72.40 | 68.33 | 64.31 | 61.07 | 58.11 | 56.23 | 54.07 | 52.13 |
| C-FSCIL (Hersche et al., 2022) | 77.47 | 72.40 | 67.47 | 63.25 | 59.84 | 56.95 | 54.42 | 52.47 | 50.47 |
| CLOM (Zou et al., 2022) | 74.07 | 70.06 | 65.90 | 61.92 | 58.83 | 55.75 | 53.62 | 51.55 | 49.56 |
| LIMIT (Zhou et al., 2023) | 73.81 | 72.09 | 67.87 | 63.89 | 60.70 | 57.77 | 55.67 | 53.52 | 51.23 |
| Bidist (Zhao et al., 2023) | 69.33 | 65.52 | 61.83 | 58.04 | 55.09 | 52.14 | 49.91 | 48.36 | 46.05 |
| TEEN (Wang et al., 2023b) | 78.92 | 72.32 | 68.16 | 64.43 | 61.19 | 58.48 | 56.11 | 54.03 | 51.87 |
| SAVC (Song et al., 2023) | 78.47 | 72.31 | 67.49 | 62.41 | 59.10 | 55.95 | 53.81 | 51.54 | 49.16 |
| OSHHG (Cui et al., 2024) | 63.55 | 62.88 | 61.05 | 58.13 | 55.68 | 54.59 | 52.93 | 50.39 | 49.48 |
| EHS (Deng & Xiang, 2024) | 73.98 | 70.11 | 66.66 | 62.75 | 60.11 | 57.33 | 55.59 | 53.75 | 51.59 |
| MICS (Kim et al., 2024) | 78.20 | 73.51 | 68.91 | 65.01 | 62.14 | 59.29 | 57.27 | 55.08 | 52.86 |
| DyCR (Pan et al., 2025) | 75.73 | 73.29 | 68.71 | 64.80 | 62.11 | 59.25 | 56.70 | 54.56 | 52.24 |
| CSR (Hu et al., 2025b) | 79.37 | 73.79 | 69.41 | 65.12 | 61.63 | 58.48 | 56.58 | 54.70 | 52.58 |
| D2A (Zhao et al., 2025) | 77.13 | 71.91 | 69.16 | 65.23 | 62.25 | 59.60 | 57.89 | 56.23 | 53.96 |
| PGLS (Hu et al., 2025a) | 82.75 | 77.60 | 73.14 | 69.28 | 65.61 | 62.49 | 60.38 | 57.93 | 55.54 |
| Ours | **83.10** | **78.72** | **74.13** | **69.60** | **66.79** | **63.84** | **62.26** | **60.68** | **58.62** |

Table 9: Comparison with the state-of-the-art methods on CIFAR100 dataset.

| Method | Inc acc. in each session (%) | | | | | | | |
|---|---|---|---|---|---|---|---|---|
| | 1 | 2 | 3 | 4 | 5 | 6 | 7 | 8 |
| CEC (Zhang et al., 2021) | 27.20 | 23.80 | 20.07 | 19.50 | 20.56 | 20.57 | 20.11 | 19.50 |
| FACT (Zhou et al., 2022) | 31.20 | 28.40 | 24.33 | 23.00 | 22.44 | 23.43 | 22.29 | 21.55 |
| C-FSCIL (Hersche et al., 2022) | 18.00 | 13.60 | 13.00 | 12.65 | 15.36 | 16.30 | 15.74 | 16.73 |
| CLOM (Zou et al., 2022) | 24.00 | 19.90 | 17.40 | 16.85 | 16.20 | 17.07 | 17.00 | 17.10 |
| LIMIT (Zhou et al., 2023) | 26.80 | 24.40 | 21.07 | 20.85 | 20.64 | 21.40 | 20.86 | 19.93 |
| Bidist (Zhao et al., 2023) | 36.80 | 31.20 | 28.80 | 25.75 | 24.36 | 22.87 | 22.43 | 20.35 |
| TEEN (Wang et al., 2023b) | 33.00 | 30.10 | 27.33 | 26.00 | 25.56 | 25.00 | 24.20 | 23.60 |
| SAVC (Song et al., 2023) | 30.60 | 23.50 | 20.13 | 20.15 | 21.36 | 22.43 | 21.77 | 21.15 |
| OSHHG (Cui et al., 2024) | 39.26 | 32.26 | 31.31 | 30.12 | 26.95 | 24.08 | 24.51 | 25.89 |
| MICS (Kim et al., 2024) | 26.80 | 22.60 | 20.27 | 21.55 | 22.12 | 23.03 | 22.26 | 21.80 |
| DyCR (Pan et al., 2025) | 31.60 | 27.00 | 22.87 | 21.85 | 21.36 | 21.30 | 20.43 | 20.22 |
| CSR (Hu et al., 2025b) | 34.40 | 26.90 | 22.20 | 21.05 | 22.16 | 23.17 | 23.46 | 22.98 |
| D2A (Zhao et al., 2025) | 34.00 | 29.70 | 26.73 | 27.20 | 28.04 | 29.47 | 29.17 | 27.70 |
| Ours | **56.40** | **48.20** | **41.00** | **40.00** | **39.40** | **41.20** | **40.26** | **39.20** |

Table 10: Comparison with the state-of-the-art methods on CUB200 dataset.

| Method | Overall acc. in each session (%) | | | | | | | | | | |
|---|---|---|---|---|---|---|---|---|---|---|---|
| | 0 | 1 | 2 | 3 | 4 | 5 | 6 | 7 | 8 | 9 | 10 |
| TOPIC (Tao et al., 2020) | 68.68 | 62.49 | 54.81 | 49.99 | 45.25 | 41.40 | 38.35 | 35.36 | 32.22 | 28.31 | 26.26 |
| CEC (Zhang et al., 2021) | 75.85 | 71.94 | 68.50 | 63.50 | 62.43 | 58.27 | 57.73 | 55.81 | 54.83 | 53.52 | 52.28 |
| FACT (Zhou et al., 2022) | 75.90 | 73.23 | 70.84 | 66.13 | 65.56 | 62.15 | 61.74 | 59.83 | 58.41 | 57.89 | 56.94 |
| CLOM (Zou et al., 2022) | 79.57 | 76.07 | 72.94 | 69.82 | 67.80 | 65.56 | 63.94 | 62.59 | 60.62 | 60.34 | 59.58 |
| LIMIT (Zhou et al., 2023) | 75.89 | 73.55 | 71.99 | 68.14 | 67.42 | 63.61 | 62.40 | 61.35 | 59.91 | 58.66 | 57.41 |
| Bidist (Zhao et al., 2023) | 75.91 | 72.32 | 70.12 | 66.04 | 64.37 | 62.18 | 60.71 | 59.62 | 57.41 | 56.68 | 55.94 |
| TEEN (Wang et al., 2023b) | 77.26 | 76.13 | 72.81 | 68.16 | 67.77 | 64.40 | 63.25 | 62.29 | 61.19 | 60.32 | 59.31 |
| SAVC (Song et al., 2023) | 81.85 | 77.92 | 74.95 | 70.21 | 69.96 | **67.02** | **66.16** | **65.30** | **63.84** | 63.15 | 62.50 |
| OSHHG (Cui et al., 2024) | 63.20 | 62.61 | 59.83 | 56.82 | 55.07 | 53.06 | 51.56 | 50.05 | 47.50 | 46.82 | 45.87 |
| M2SD (Lin et al., 2024) | 81.49 | 76.67 | 73.58 | 68.77 | 68.73 | 65.78 | 64.73 | 64.03 | 62.70 | 62.09 | 60.96 |
| MICS (Kim et al., 2024) | 78.81 | 75.31 | 72.25 | 68.72 | 67.43 | 65.28 | 64.61 | 63.39 | 61.77 | 61.76 | 61.29 |
| DyCR (Pan et al., 2025) | 77.50 | 74.73 | 71.69 | 67.01 | 66.59 | 63.43 | 62.66 | 61.69 | 60.57 | 59.69 | 58.46 |
| CSR (Hu et al., 2025b) | **81.90** | 77.66 | 74.82 | 70.27 | 69.71 | 66.82 | 66.36 | 64.98 | 63.35 | 62.72 | 61.80 |
| D2A (Zhao et al., 2025) | 81.46 | **78.43** | **75.28** | 71.19 | 69.71 | 66.67 | 65.37 | 64.28 | 62.49 | 62.40 | 61.46 |
| PGLS (Hu et al., 2025a) | 79.59 | 76.63 | 73.71 | 69.01 | 68.28 | 65.34 | 64.52 | 63.57 | 62.10 | 61.69 | 60.62 |
| Ours | 80.76 | 77.60 | 75.01 | **71.43** | **70.08** | 66.76 | 66.10 | 65.02 | 63.43 | **63.41** | **62.89** |

Table 11: Comparison with the state-of-the-art methods on CUB200 dataset.

| Method | Inc acc. in each session (%) | | | | | | | | | |
|---|---|---|---|---|---|---|---|---|---|---|
| | 1 | 2 | 3 | 4 | 5 | 6 | 7 | 8 | 9 | 10 |
| CEC (Zhang et al., 2021) | 40.14 | 37.45 | 30.90 | 33.82 | 31.32 | 32.75 | 32.91 | 31.46 | 33.22 | 33.02 |
| FACT (Zhou et al., 2022) | 52.33 | 47.02 | 37.61 | 39.70 | 37.59 | 39.25 | 39.79 | 38.25 | 40.14 | 39.74 |
| CLOM (Zou et al., 2022) | 37.99 | 37.44 | 32.94 | 36.49 | 35.70 | 37.97 | 39.08 | 38.69 | 40.60 | 40.67 |
| LIMIT (Zhou et al., 2023) | 51.21 | 45.40 | 37.57 | 39.26 | 37.55 | 38.27 | 39.92 | 38.14 | 39.38 | 41.83 |
| Bidist (Zhao et al., 2023) | 55.20 | 46.11 | 38.19 | 41.67 | 37.62 | 38.11 | 39.28 | 36.81 | 39.73 | 39.83 |
| TEEN (Wang et al., 2023b) | 57.71 | 52.56 | 45.55 | 47.41 | 44.97 | 46.10 | 45.62 | 43.54 | 44.94 | 44.84 |
| SAVC (Song et al., 2023) | 60.57 | 53.39 | 44.18 | 46.28 | 43.23 | 45.11 | 45.67 | 44.16 | 45.82 | 45.79 |
| OSHHG (Cui et al., 2024) | 56.81 | 43.17 | 35.16 | 34.99 | 33.03 | 32.41 | 31.51 | 28.13 | 28.86 | 28.73 |
| MICS (Kim et al., 2024) | 48.75 | 46.29 | 40.51 | 44.24 | 43.43 | 45.66 | 45.82 | 44.66 | 46.77 | 47.47 |
| DyCR (Pan et al., 2025) | 46.24 | 42.76 | 37.64 | 41.64 | 40.41 | 42.01 | 42.08 | 39.51 | 41.04 | 41.90 |
| CSR (Hu et al., 2025b) | 54.84 | 52.30 | 44.56 | 46.56 | 44.05 | 45.54 | 46.17 | 44.79 | 46.35 | 46.35 |
| D2A (Zhao et al., 2025) | **61.65** | 55.15 | 45.59 | 48.07 | 44.87 | 46.48 | 47.13 | 45.52 | 47.45 | 47.43 |
| Ours | 60.93 | **57.06** | **48.44** | **50.89** | **47.14** | **48.94** | **49.20** | **47.87** | **49.62** | **50.15** |

