# OpenReview forum: "Hierarchical Filtering and Refinement Classification for Few-Shot Class-Incremental Learning"
_TMLR — Accepted by TMLR_

### Review · Reviewer_LYSQ · 2026-02-11

**Summary Of Contributions:**

This paper studies few-shot class-incremental learning (FSCIL). The authors argue that a common “train in base, freeze the backbone, add novel prototypes” paradigm yields good overall accuracy but poor incremental-class accuracy, partly because treating all classes under one unified classification objective makes it hard to satisfy both base- and incremental-class requirements. To address this, the paper proposes Hierarchical Filtering and Refinement Classification (HFRC). It first decomposes the FSCIL objective (via Bayes’ rule) into (i) inter-group classification (base vs. incremental) and (ii) intra-group classification (within the predicted group). Method-wise, HFRC introduces a Selection & Reorganization (SR) module after the feature extractor to induce two feature types: transferable features (before SR) intended to generalize to future incremental classes, and class-specific features (after SR) intended to better separate base classes. At inference, HFRC performs a two-stage decision: it uses transferable features to do a first-pass prototype classification (also acting as a coarse base-vs-incremental filter), and if the prediction falls into the base set, it “refines” using class-specific features to better separate base classes and resolve base-vs-incremental confusions. The paper also proposes a classifier adjustment that shifts base-class prototypes away from incremental prototypes using an accumulated direction term and a scale hyperparameter. Experiments on miniImageNet, CIFAR100, and CUB200 with a ResNet18 backbone show improved overall accuracy and notably better incremental-class accuracy.

**Additional Comments:**

NA

**Audience:**

Yes

**Audience Explanation:**

NA

**Claims And Evidence:**

Yes

**Claims Explanation:**

NA

**Requested Changes:**

1.	The definitions of “class-specific” vs. “transferable” features and why a simple SR  reliably induces this hierarchy feel heuristic, and the paper would benefit from a clearer mechanism-level explanation that SR causally encourages transferability rather than just changing representation capacity.
2.	The refinement step is triggered based on the first-pass prediction, which can create failure modes where an incremental sample misrouted early never gets corrected (or vice versa). A more thorough analysis (e.g., routing confusion matrices, confidence-based gating variants, or “what fraction of errors are due to mis-filtering vs. intra-group confusion”) would strengthen the claim that the decomposition truly resolves the base–incremental tension.
3.	The paper claims incremental stages are efficient and inference overhead is negligible, but it does not provide concrete runtime/memory numbers; similarly, classifier adjustment uses (\gamma) and is only lightly analyzed. Please add (i) wall-clock and memory comparisons vs. representative baselines, and (ii) broader sensitivity across datasets/protocol variants (shots/ways, different base splits, multiple seeds with variance), especially since results can be sensitive to augmentation and evaluation details.

---

> ### Author Response · Authors · 2026-03-14
> **Author Response (1/2)**
>
> We thank the reviewer for the valuable feedback and comments. Below, we provide a point-by-point response and outline the main corresponding revisions made to the manuscript.
>
> **Responses to Requested Changes 1:**
>
> We thank the reviewer for the insightful comment. To provide a clearer mechanism-level explanation of how the SR module encourages transferability, we have added a “Gradient Analysis” subsection in Section 3.3 (page 5).
>
> In Section 3.3, we explicitly defines the activation behaviors of class-specific versus transferable features. This mechanism is experimentally visualized in Section 4.4.1: without $SR$, the backbone feature $g(\mathbf{x})$ is concentrated on a small number of dimensions, heavily biased toward base classes; after introducing $SR$, $SR$ absorbs discriminative pressure, “freeing” the backbone to preserve more broadly activated semantic features. Thus, $SR$ does not merely change representation capacity but redistributes gradient pressure by shifting the discriminative task downstream.
>
> To further quantify how discriminative gradients are distributed across backbone feature dimensions, we analyze the gradient behavior from the perspective of the optimization objective in the newly added “Gradient Analysis” subsection of Section 3.3.
>
> >During base training, the classification objective can be written as
> $\mathcal{L}\big( \{ \eta_ c^\top \tilde{g}(x) \}_{c \in \mathcal{C}_0}, y \big)$, where $\mathcal{S}(\cdot,\cdot)$ denotes the cosine similarity function.
> >
> >For $\tilde{g}(x)$, the gradient direction is strongly determined by the classifier parameters $\eta _ c$, i.e.,
> \begin{equation}
> \frac{\partial \mathcal{L}}{\partial \tilde{g}(\mathbf{x})}=\sum_{c \in \mathcal{C} _ 0}
> \frac{\partial \mathcal{L}}{\partial \mathcal{S}(\eta _ c, \tilde{g}(\mathbf{x}))}
> \frac{\partial \mathcal{S}(\eta _ c, \tilde{g}(\mathbf{x}))}{\partial \tilde{g}(\mathbf{x})}
> \in\mathrm{span} \lbrace \eta _ c \mid c \in \mathcal{C} _ 0 \rbrace.
> \end{equation}
> >
> >Since the number of base classes is limited, the subspace spanned by $\{ \eta _ c \}$ typically has much lower dimensionality than the full feature space. Consequently, there exists a subset of feature dimensions $D_{\text{base}} \subset  \lbrace 1,\dots,D  \rbrace $ with $|D_{\text{base}}| \ll D$ such that
> \begin{equation}
> \left|
> \frac{\partial \mathcal{L}}{\partial \tilde{g}_ d(x)}
> \right|
> \approx 0,
> \quad
> \forall d \notin D_{\text{base}},
> \end{equation}
> implying that only a small number of feature channels are strongly activated by base class supervision.
> >
> >For the backbone feature $g(\mathbf{x})$, applying the chain rule yields
> >
> >\begin{equation}
> \frac{\partial \mathcal{L}}{\partial g(\mathbf{x})} = \frac{\partial \tilde{g}(\mathbf{x})}{\partial g(\mathbf{x})}
> \frac{\partial \mathcal{L}}{\partial \tilde{g}(\mathbf{x})},
> \end{equation}
> >
> >where $\frac{\partial \tilde{g}(\mathbf{x})}{\partial g(\mathbf{x})}$ denotes the Jacobian of the SR module in the backward pass.
> >
> >From a forward perspective, the SR parameters are directly optimized to absorb the discriminative gradients required for base class classification.
> From a backward perspective, the SR module acts as a gradient reallocation mechanism, redistributing the gradients propagated back to the backbone through its Jacobian.
> As a result, excessive optimization pressure toward a small set of class-specific directions is alleviated, encouraging the backbone to preserve more transferable feature representations.
>
>
> **Responses to Requested Changes 2:**
>
> Thank you for this helpful suggestion. In Table 4 of Section 4.4.3, we compare several alternative routing strategies, all of which are less effective than our proposed routing mechanism. To further provide an intuitive demonstration that the proposed decomposition truly resolves the base-incremental tension, we have additionally included routing confusion matrices in Figure 5 and corresponding analysis in Section 4.4.3 (page 11) of the revised manuscript.
>
> Based on the observations from the confusion matrices, we find that under transferable features, almost no base class samples are misclassified as incremental classes, i.e., low mis-filtering. Meanwhile, the dominant source of ambiguity lies within the base-group branch after filtering, i.e., high base-group confusion. Consequently, applying class-specific features for refinement (**RC**) only to samples filtered into the base group (**FC**) is particularly effective. These findings are also consistent with the ablation results reported in Table 3 of the manuscript, where both the filtering (**FC**) and the refinement (**RC**) consistently improve classification accuracy.

---

> ### Author Response · Authors · 2026-03-14
> **Author Response (2/2)**
>
> **Responses to Requested Changes 3:**
>
> (i) **Runtime and memory analysis**
>
> We thank the reviewer for raising this important point. To provide a clearer evaluation of incremental efficiency and strengthen the empirical evidence, we have added comparisons experiments of training time, inference time, and memory consumption with recent and state-of-the-art methods, along with the corresponding analysis in Section 4.4.6 (page 13) of the revised manuscript.
>
> As shown in Table 6 of the manuscript, the results are consistent with the algorithmic analysis presented in Section 4.1.2 of the paper. During the incremental session, the computation only involves a simple classifier update, without any backpropagation-based optimization. As a result, our method significantly reduces the training time in the incremental session compared with recent approaches.
> Furthermore, during inference, our method only introduces a lightweight forward computation and similarity calculation. Therefore, it has negligible impact on inference time, and is even slightly faster than methods that also introduce additional strategy during inference (e.g., D2A). Moreover, our method requires a comparable amount of memory to other methods during training. For convenience, we also provide the results below:
>
> | Method | Training time (s) | Inference time (ms) | Memory (MB) |
> |--------|:-----------------:|:-------------------:|:-----------:|
> | SAVC   | 22.09 | 1.06 | 299.68 |
> | CSR    | 43.37 | 1.09 | 306.05 |
> | D2A    | 22.05 | 1.27 | 300.18 |
> | Ours   | 2.94  | 1.08 | 280.15 |
>
> (ii) **Broader sensitivity across datasets and protocol variants**
>
> To further demonstrate the robustness of our method to different datasets and protocol variants, we have conducted additional experiments under several protocol variations, including different numbers of shots and ways, different base class splits, and multiple random seeds across datasets. The corresponding experiments and analyses are included in Sections 4.4.4 and 4.4.5 (page 12) of the revised manuscript.
>
> As shown in Figure 7 and Figure 8 of the manuscript, changing the number of shots or ways exhibits stable and consistent trends across all datasets. For convenience, we also summarize the key observations below. As expected, performance steadily improves as the number of shots increases. When varying the number of ways, the performance remains similar for sessions with the same number of seen classes, indicating that our method is insensitive to the specific way partitioning.
>
> As reported in Table 5 of the manuscript, we evaluate our method under three randomly sampled base class splits and three different random seeds. Compared with the standard setting, the results exhibit only minor variations, demonstrating stable performance. For convenience, we also provide the results below:
>
> |  | miniImageNet (Overall / Inc.) | CIFAR100 (Overall / Inc.) | CUB200 (Overall / Inc.) |
> |---|---|---|---|
> | Base splits | 70.21±0.88 / 43.71±1.16 | 68.25±0.52 / 42.61±0.85 | 69.79±0.67 / 51.69±0.94 |
> | Seeds | 70.43±0.57 / 43.95±0.82 | 68.74±0.18 / 42.99±0.31 | 69.09±0.32 / 50.50±0.74 |
>
> In addition, we further analyze the classifier adjustment parameter across all datasets in Figure 6 of the manuscript, which also shows consistent performance trends.

---

### Review · Reviewer_YuWq · 2026-02-21

**Summary Of Contributions:**

This paper studies few-shot class-incremental learning. To balance the classification accuracy between base and incremental classes, the authors propose decomposing the objective into classification between base and incremental classes and classification within each group, which is an interesting and novel idea. To address these two distinct sub-objectives, the authors introduce a new classification framework. The main idea is to learn transferable features and class-specific features and leverage them in a hierarchical filtering and refinement process, where transferable features are used for base–incremental separation and incremental classification, while class-specific features refine base-class predictions. Experimental results on real-world datasets demonstrate the effectiveness of the proposed method.

**Audience:**

Yes

**Audience Explanation:**

This paper studies class-incremental machine learning, which is an important and classical ML problem. The problem is relevant to the TMLR community, particularly researchers working on continual learning, representation learning, and few-shot learning.

**Claims And Evidence:**

Yes

**Claims Explanation:**

In terms of methodology, this paper provides a clear analysis of the objective decomposition and presents a detailed model pipeline. In the experiments section, the authors conduct extensive experiments on three datasets and compare their method with a wide range of previous approaches. I also appreciate the ablation studies, which demonstrate the effectiveness of the proposed method.

**Requested Changes:**

In the introduction, the authors say that "Based on Bayes’ theorem, we decompose the overall classification objective into three sub-objectives", but it seems that there are only two objectives. Is this a typo?

It would be great if the authors could add more explanation in the caption of Figure 1, for example, "SR() stands for Selection and Reorganization module".

---

> ### Author Response · Authors · 2026-03-14
> **Author Response**
>
> We thank the reviewer for the valuable feedback and comments. Below, we provide a point-by-point response and outline the main corresponding revisions made to the manuscript.
>
> **Responses to Requested Changes 1:**
>
> We apologize for the confusion caused by describing the two intra-group objectives together in the original manuscript. Our intention is to decompose the overall classification objective into three sub-objectives:
>
> 1. Inter-group classification between base and incremental classes, which determines whether a sample belongs to the base or incremental group. This objective is mainly addressed in the filtering stage, where transferable features perform a coarse separation between base and incremental classes, while the refinement stage further uses class-specific features to distinguish true base class samples from similar incremental samples.
>
> 2. Intra-group classification within base classes, which identifies the specific base class for samples predicted as belonging to the base group. This objective is handled in the refinement stage, where class-specific features provide stronger discriminative ability for distinguishing among base classes.
>
> 3. Intra-group classification within incremental classes, which identifies the specific incremental class for samples predicted as belonging to the incremental group. This objective is primarily achieved by the transferable features learned in hierarchical feature extraction, which improve the separability of incremental classes in the feature space.
>
> In the revised manuscript, we have carefully revised the relevant sections to explicitly enumerate and describe all three sub-objectives (page 2), and further explicitly state the formulation using the chain rule of probability, thereby improving clarity and avoiding ambiguity.
>
> **Responses to Requested Changes 2:**
>
> Thank you for this helpful suggestion. We have updated the caption of Figure 1 in the revised manuscript to provide additional clarification. Specifically, we now explicitly define terms such as $g(\cdot)$ and $SR(\cdot)$, to improve readability and ensure that all components are clearly understood by the reader. In particular, $g(\cdot)$ denotes the backbone feature extractor that produces transferable features, while $SR(\cdot)$ represents the Selection and Reorganization module that reorganizes these features to produce class-specific features.

---

### Review · Reviewer_8GMs · 2026-03-05

**Summary Of Contributions:**

This paper proposes HFRC, a classification framework for FSCIL that maintains two separate feature spaces via an SR module and applies a two-stage prototype-based classification to better balance accuracy between base and incremental classes. A prototype adjustment mechanism is additionally introduced to increase the margin between base and incremental class prototypes at inference time.

**Audience:**

Yes

**Audience Explanation:**

Yes. The performance imbalance between base and incremental classes in FSCIL is a meaningful problem, and the empirical findings of this paper are likely to be of interest to researchers working in this area.

**Claims And Evidence:**

Yes

**Claims Explanation:**

Only partially. The empirical claims are well-supported by consistent results across three benchmarks. However, several theoretical claims in the paper are overclaimed. Most notably, Equation 4 is presented as a principled decomposition grounded in Bayes' theorem, but it is simply the chain rule of probability. Similarly, the classifier adjustment is a common prototype-level operation that has been explored in prior work, and presenting it as a novel theoretical contribution is not well-justified. More broadly, a number of equations throughout the paper do not add substantial insight and appear to be included primarily to formalize relatively straightforward operations.

**Requested Changes:**

1. The paper should more clearly discuss what is new compared to prior work on prototype adjustment, and better justify the novelty of the proposed approach.

2. Equation 4 is the chain rule of probability and should not be framed as a Bayesian contribution.

3. Several equations in the paper add little value and should either be better motivated or removed.

4. Figure 1 should be revised to more clearly show how prototypes and decision boundaries change across the two stages.

---

> ### Author Response · Authors · 2026-03-14
> **Author Response (1/2)**
>
> We thank the reviewer for the valuable feedback and comments. Below, we provide a point-by-point response and outline the main corresponding revisions made to the manuscript to better align our claims with accurate, clear, and convincing supporting evidence.
>
> **Responses to Requested Changes 1:**
>
> Thank you for this helpful comment. We clarify the novelty of our approach and its difference from prior work on prototype adjustment as follows.
>
> 1. Our core idea is to decompose the overall FSCIL classification objective into multiple sub-objectives, including inter-group classification (base vs. incremental classes) and intra-group classification within each group. Motivated by this decomposition, we propose a novel approach called Hierarchical Filtering and Refinement Classification (HFRC). HFRC addresses these sub-objectives through two key components:
> - Hierarchical Feature Extraction, which learns two complementary types of features during training, transferable features that generalize well to incremental classes and class-specific features that are more discriminative for base classes.
> - Filtering and Refinement Classification, which exploits these two feature types in a hierarchical classification procedure to handle the decomposed sub-objectives during inference.
>
>
>   Therefore, the main novelty and contribution lies in the structured decomposition of the classification objective and the resulting hierarchical classification framework, rather than in prototype adjustment itself.
>
> 2. Our classifier adjustment mechanism is designed as a lightweight operation within the refinement stage. Specifically, it dynamically adjusts base class prototypes by slightly shifting them away from distribution of incremental classes. This helps the refinement classification better distinguish between base and incremental classes that are easily confused.
>
>
>    This design is fundamentally different from prior prototype adjustment methods. For example, CEC aggregates information from all classifiers in the graph and fuses it with the original classifier, while TEEN calibrates incremental class prototypes by combining them with base prototypes. In contrast, our method directly shifts base prototypes away from the distribution of incremental classes, which helps balance the refinement decision boundary and mitigate its bias toward base classes, instead of calibrating prototypes using cross-class information.
>
> In the revised manuscript, we have revised the introduction in Section 1 to better highlight the main novelty and contributions of our approach (page 2) and updated the related work in Section 2.3 to compare our method with prototype adjustment approaches (page 3).
>
> **Responses to Requested Changes 2:**
>
> We agree with the reviewer that Equation (4) is mathematically equivalent to the chain rule of probability. Our intention was not to claim a Bayesian theoretical contribution, but to provide an intuitive probabilistic interpretation that motivates the hierarchical Filtering and Refinement Classification.
>
> In the revised manuscript, to avoid confusion, we have replaced “based on Bayes’ theorem” with “using the chain rule of probability” (page 2, page 5) and clarified that the equation is used to motivate the hierarchical classification procedure (page 5).

---

> ### Author Response · Authors · 2026-03-14
> **Author Response (2/2)**
>
> **Responses to Requested Changes 3:**
>
> Thank you for this helpful suggestion. We have removed several equations that describe standard operations in the previous manuscript, including the base training objective (Equation (1)) and the base session classification rule (Equation (6)).
>
> After the revision, only the equations that are closely related to the core idea and method are retained, and each equation is clearly motivated. Specifically, in the revised manuscript, Equation (1) and Equation (4) are used to define the two types of features employed in our framework, Equations (2)-(3) present the probabilistic decomposition that motivates the hierarchical classification design, Equations (8)-(9) define the hierarchical classification procedure, Equations (10)-(11) describe the classifier adjustment mechanism, and Equations (5)-(7) explain how the SR module encourages transferable feature learning. Furthermore, since Equation (2) is referenced later for feature definitions, we have added additional explanation immediately after Equation (2) to clarify its role and better motivate this formulation (page 4).
>
> **Responses to Requested Changes 4:**
>
> Thank you for this helpful suggestion. In Figure 1, we use solid and hollow pentagrams to represent the prototypes defined in the two different feature spaces used in the filtering and refinement stages, respectively. In the filtering stage, classification is performed in the transferable feature space, which enables correct recognition of the majority of incremental class test samples. In the refinement stage, all test samples that are routed to the base class group are further classified using class-specific features, which resolves most base class samples as well as incremental samples that are prone to confusion with base classes. The final prediction is obtained by merging the classification results from the two feature spaces.
>
> To more clearly illustrate how prototypes and decision boundaries change across the two stages, we have revised Figure 1 (page4) in the following ways in the revised manuscript:
>
> 1. We have explicitly distinguished and separately annotated the prototypes of base classes and incremental classes, highlighting how the corresponding prototypes differ between the two stages.
>
> 2. We have added the decision boundaries among base classes in the first stage and refined their positions to better reflect the classification process, making the contrast between the decision boundaries in the filtering stage and the refinement stage more explicit.

---

### Author Response · Authors · 2026-03-14
**Response Summary**

We would like to express our sincere appreciation to the editors and reviewers for their time, effort, and insightful comments on our manuscript. In response to the constructive feedback, we have carefully addressed all the raised concerns and thoroughly revised the manuscript accordingly. The revised version of the manuscript has been uploaded, and all revisions and additions are highlighted in blue.

The main revisions and responses are summarized as follows:

- We have revised the introduction in Section 1 to better highlight the main novelty and contributions of our approach, and updated the related work in Section 2.3 to compare our method with prototype adjustment approaches.
- We have explicitly stated that the formulation uses the chain rule of probability and enumerated the three sub-objectives in the relevant sections to improve clarity and avoid ambiguity.
- We have removed several equations and provided clearer motivation for the remaining ones. We have also revised Figure 1 to more clearly illustrate how prototypes and decision boundaries change across the two stages, and updated its caption to provide additional clarification.
- We have added a “Gradient Analysis” subsection in Section 3.3 to provide a clearer mechanism-level explanation of how the SR module encourages transferability.
- We have added routing confusion matrices in Figure 5 and the corresponding analysis in Section 4.4.3 to provide a more intuitive demonstration that the proposed decomposition resolves the base–incremental tension.
- We have added runtime and memory analyses in Section 4.4.6, as well as broader sensitivity analyses across datasets and protocol variants in Sections 4.4.4 and 4.4.5 to strengthen the empirical evaluation and further demonstrate the robustness of our method.

---

### Decision · Action_Editor_XVGm · 2026-04-24

**Recommendation:** Accept as is

**Audience:**

Yes

**Audience Explanation:**

The paper addresses the performance imbalance between base and incremental classes in FSCIL, which is a relevant and timely problem in continual learning and few-shot learning. The empirical results and insights from the hierarchical decomposition and SR module are likely to interest researchers in machine learning.

**Claims And Evidence:**

Yes

**Claims Explanation:**

The submission provides strong empirical support for its claims. The hierarchical filtering and refinement classification framework is validated across multiple FSCIL benchmarks with ablation studies, confusion matrices, and sensitivity analysis. While some theoretical claims (e.g., Equation 4) were initially overstated, the authors have clarified these points in the revision. Overall, the empirical evidence convincingly supports the main claims regarding improved incremental class performance.